# *ALDH1A3* Regulates Cellular Senescence and Senescence-Associated Secretome in Prostate Cancer

**DOI:** 10.3390/cancers17071184

**Published:** 2025-03-31

**Authors:** Sen Wang, Lin Wang, Yu Zhao

**Affiliations:** 1Tianjin Key Laboratory of Radiation Medicine and Molecular Nuclear Medicine, Institute of Radiation Medicine, Chinese Academy of Medical Sciences and Peking Union Medical College, Tianjin 300192, China; wangsen@irm-cams.ac.cn; 2School of Medicine, Nankai University, 94 Weijin Road, Tianjin 300071, China; wl1194079931@163.com

**Keywords:** *ALDH1A3*, cellular senescence, SASP, cGAS–STING, radiotherapy

## Abstract

The aim of this study was to investigate the regulatory role of *ALDH1A3* in radiation-induced aging and age-related secretory phenotype (SASP). While effectively controlling tumor growth, radiotherapy can also trigger complex cellular reactions, such as cell senescence, which has an impact on tumor progression. The aging phenotype was accelerated by knocking down *ALDH1A3*, and SASP secretion was inhibited by cGAS-STING pathway.

## 1. Introduction

Prostate cancer is one of the most common male malignant tumors, second only to lung cancer [1]. Radiotherapy is one of the important means in the treatment of prostate cancer patients, which can effectively control tumor growth, especially for local cancer [2,3]. At different stages, patients may receive radiotherapy as primary treatment or as an adjunct after surgery [4,5]. In a non-invasive way, radiotherapy is able to damage the DNA of cancer cells and induce senescence or apoptosis, thus achieving tumor control [6]. However, up to 30% of prostate cancer (PCa) patients who undergo surgery and radiation therapy experience recurrence and metastasis. This may be due to the complex cellular responses triggered by radiation, which can lead to irreversible cell cycle arrest and cellular senescence.

Cellular senescence is related to a variety of biological functions and processes, and has both beneficial and harmful effects on the human body. A characteristic of cellular senescence is the accumulation of damaged molecules and genomic instability [7]. Cellular senescence has a protective effect by reducing the risk of cancer through the prevention of damaged cells from dividing and proliferating, and by accelerating immune clearance [8]. Another highly representative feature of this process is the emergence of the senescent-associated secretory phenotype (SASP) [9]. SASP includes a variety of pro-inflammatory cytokines, chemokines, and growth factors, which can significantly alter the tumor microenvironment and even drive tumor growth and metastasis under certain conditions [10,11]. However, it is important to note that the presence of a single marker is not sufficient to define an aging state, as in some cases, almost all currently discovered markers may be absent in senescent cells or present in non-senescent cells.

While the SASP helps to motivate the immune system to clear senescent cells, it also has a double-edged effect: a sustained inflammatory response and immune escape can fuel cancer progression [12]. Therefore, it is of great clinical significance to study the regulatory mechanism of cellular senescence and its relationship with the tumor microenvironment. In this process, the cGAS–STING pathway, as a key immune response regulation mechanism, activates STING by sensing DNA fragments in the cytoplasm, and then triggers a series of pro-inflammatory reactions, which is closely related to the maintenance of SASP [13,14,15].

*ALDH1A3*, a member of the aldehyde dehydrogenase family, is believed to play an important role in glycolysis and cancer cell metabolism [16,17]. Recent studies have found that the high expression of *ALDH1A3* in some tumors is closely related to the maintenance and radiotherapy resistance of cancer stem cells (CSCs) [18]. It not only regulates intracellular metabolism, but also may enhance the DNA repair ability of cancer cells by interacting with PKM2, promoting their resistance to radiotherapy [19].

In this study, we investigated the effect of *ALDH1A3* on cellular senescence. Through in vitro senescence and tumor growth models, we determined that *ALDH1A3* knockdown accelerates the senescent-like phenotype process and that *ALDH1A3* plays an important role in the induction of SASP. Further analysis shows that the performance of SASP depends on the expression of *ALDH1A3*. In addition, we reveal the mechanism by which *ALDH1A3* plays a role in this process and provide foundational evidence that *ALDH1A3* can be used as a potential drug target to regulate the behavior of senescent cells and improve the effectiveness of radiotherapy. This provides molecular mechanistic insights into future therapeutic strategies.

## 2. Materials and Methods

### 2.1. Cell Culture

HEK 293T and PC3 cells were purchased from Wuhan Promoter Life Science & Technology Co., Ltd. (Wuhan, China), and C4-2 cells were obtained from Tianjin Hamosi Technology Development Co., Ltd. (Tianjin, China). All cell lines were cultured at 37 °C in an atmosphere of 5% CO_2_ using DMEM or 1640 medium containing 10% fetal bovine serum (FBS) (Gibco, Thermo Fisher Scientific, Shanghai, China) and 1% penicillin/streptomycin. When cell confluence reached 70–80%, they were subjected to ionizing radiation treatment or lentiviral transduction. 

### 2.2. RNA Sequencing

The irradiated or non-irradiated PC3 cells were sent to Novogene for sequencing. Each sample had three replicates.

### 2.3. Mass Spectrometry

PC3 cells that were non-irradiated, irradiated with 20 Gy for 5 days, and irradiated with 20 Gy for 7 days were lysed with RIPA lysis buffer (Solarbio, Beijing, China) at 4 °C for 20 min. The protein samples were then separated by SDS-PAGE, and the gel strips were sent to Novogene for mass spectrometry analysis. Each sample had three replicates.

### 2.4. Real-Time qPCR

Trizol was used to extract RNA from cells. After RNA extraction, the quantity and quality of the RNA were analyzed using the Qnano spectrophotometry method from Yeasen. A reverse transcription reaction was performed using 1 μg of total RNA with Takara’s reverse transcription kit (RR047A). The resulting cDNA was subjected to qPCR analysis using Hieff UNICON^®^ Universal Blue qPCR SYBR Green Master Mix (Yeasen, Cat #11184ES08, Shanghai, China), and quantification was performed using Yeasen 80520ES03. In all experimental replicates, all expression levels were normalized to GAPDH. The primers used were as follows:

*CDKN2A Forward* (5′-CTTCCTGGACACGCTGGTGG-3′)

*CDKN2A Reverse* (5′-TCGGGGATGTCTGAGGGACC-3′)

*CDKN1A Forward* (5′-TGTCCGTCAGAACCCATGC-3′)

*CDKN1A Reverse* (5′-AAAGTCGAAGTTCCATCGCTC-3′)

*GAPDH Forward* (5′-GGAGCGAGATCCCTCCAAAAT-3′)

*GAPDH Reverse* (5′-GGCTGTTGTCATACTTCTCATGG-3′)

### 2.5. Data Collection and Analysis of Differential Expression Genes

To obtain gene expression and clinical information from patients, we utilized The Cancer Genome Atlas (TCGA) database to collect data from patients with PRAD. Subsequently, we employed the R package “edgeR” to conduct differentially expressed genes (DEGs) analysis on these datasets. We set |logFC| ≥ 1 and adj. *p* < 0.05 as the selection criteria to define the DEGs in the database.

### 2.6. Drug Sensitivity

We explored the predictive value of *ALDH1A3* expression in relation to the efficacy of immunotherapy and chemotherapy. Tumor immune dysfunction and exclusion (TIDE) was used to predict patients’ responses to immunotherapy. The “oncoPredict” package in R was applied to estimate drug sensitivity.

### 2.7. SA-β-Gal Staining

Senescent cells were detected using the Senescence β-Galactosidase Staining Kit (Beyotime, Cat# C0602, Shanghai, China) according to the manufacturer’s protocol.

### 2.8. Western Blot

Western blot was performed using total protein extracts. The following antibodies were used: cGAS (Cell Signaling Technology, Cat# 15102, Shanghai, China), STING (Cell Signaling Technology, Cat# 13647, Shanghai, China), Phospho-STING (Cell Signaling Technology, Cat# 19781, Shanghai, China), TBK1 (Cell Signaling Technology, Cat# 3504, Shanghai, China), Phospho-TBK1 (Cell Signaling Technology, Cat# 5483, Shanghai, China), IRF-3 (Cell Signaling Technology, Cat# 4302, Shanghai, China), STAT2 (Invitrogen, Cat# 44-362G, Thermo Fisher Scientific, Shanghai, China), p65 (Cell Signaling Technology, Cat# 8242, Shanghai, China), Beta Actin (Proteintech, Cat# 66009-1-lg, Wuhan, China), GAPDH (Proteintech, Cat# 60004-1-lg, Wuhan, China), *ALDH1A3* (Proteintech, Cat# 25167-1-AP, Wuhan, China), and p16 (Santa Cruz, Cat# sc-9968, Santa Cruz, CA, USA).

### 2.9. Flow Cytometry

The cells to be tested were collected and washed once with PBS. The cells were then resuspended in PBS and MHC I antibody (Biolegend, San Diego, CA, USA) was added to the suspension at a ratio of 5 μL per 1 million cells. Subsequently, the cell suspension was incubated at 4 °C for 30 min, and then washed twice with PBS. The cells were fixed with 4% paraformaldehyde for further analysis.

### 2.10. Lentiviral Vector Production and Transduction

The plasmids psPAX2, pMD2.G, and lentiCRISPRv2 were purchased from Addgene (Cambridge, MA, USA). The purchased lentiCRISPRv2 plasmid was digested with the BSMBI restriction enzyme (Yeasen, Cat#15203RS80), and the synthesized sgRNA was ligated into the digested lentiCRISPRv2 using T4 ligase (Takara, Shanghai, China) to obtain the lentiCRISPRv2-sg*ALDH1A3* plasmid. HEK 293T cells were transfected with psPAX2, pMD2.G, and either lentiCRISPRv2 or lentiCRISPRv2-sg*ALDH1A3*. After 72 h, the supernatant containing lentiviral particles was collected. The lentivirus was directly added to the culture medium and used to transduce PC3 cells with 8 μg/μL polybrene.

### 2.11. SgRNA

sg*ALDH1A3*-1: 5′-TAGTCTGCGGCGCACCGGCT-3′

sg*ALDH1A3*-2: 5′-TTCCACGGCCCCGTTAGCGG-3′

sg*ALDH1A3*-3: 5′-CATATTTGCTCCCGAGTTGA-3′

### 2.12. Transwell Cancer Growth Experiments

PC3 cells transduced with the virus were seeded onto transwell permeable supports (Corning, Cat# 3470, New York, NY, USA). When they reached 70–80% confluence, the supports were placed into a 24-well plate seeded with PC3 cells. Cell viability was measured every 24 h using the Cell Counting Kit-8 (CCK-8) (Biosharp, Cat# BS350B, Shanghai, China).

### 2.13. Transwell In Vitro Scratch Experiments

A transwell plate was prepared as described above. PC3 cells were seeded in a 6-well plate and cultured to confluence. A scratch was made using a 200 µL pipette tip, and debris was removed by washing the cells with a culture medium. The transwell plate was then placed into the culture plate, and scratch wound images were captured under the microscope 24 h later.

### 2.14. Statistical Analysis

All statistical analyses and graphical visualizations were performed in R (version 4.4.1). Continuous variables between groups were compared using the Student’s *t*-test or the Wilcoxon rank-sum test. *p*-value < 0.05 was considered statistically significant (two-tailed).

## 3. Results

### 3.1. Short-Term and Long-Term Effects of Irradiation on Gene Expression in Cells

In order to study the effect of irradiation on cell gene expression, we used an irradiation dose of 5 Gy in the experiment, which is a common dose to study the effect of cell irradiation. Irradiation with 5 Gy can trigger a significant biological response without causing extensive cell death and is widely used to simulate moderate doses of radiotherapy. Through this dose of irradiation, we can observe a series of reactions such as DNA damage repair, gene expression regulation, and immune response, providing a scientific basis for the study of irradiation resistance, stress response, and potential immune escape mechanism of cancer cells.

In this study, we compared the RNA-seq results of PC3 cells unirradiated (control), 24 h after 5 Gy irradiation, and 96 h after 5 Gy irradiation (Appendix A). The results showed that irradiation had a significant effect on gene expression in PC3 cells. Figure 1A shows the overall differences in gene expression among the experimental groups, further confirming the significant influence of short- and long-term effects of irradiation on gene expression. We found different effects of short-term (24 h) and long-term (96 h) effects of irradiation on cellular gene expression in water. Some genes are activated or inhibited rapidly in the short term, while others show delayed expression changes over time. This phenomenon suggests that irradiation not only has immediate effects on cells, but may also induce long-lasting changes in gene expression.

We further compared the differential genes between 96 h and control groups and between 96 h and 24 h after irradiation. |logFC| ≥ 1 and adj. *p* < 0.05 were used as truncation criteria. The left side of Figure 1B shows the number of up-regulated and down-regulated genes compared to the control group after 96 h of irradiation, while the right side of Figure 1B compares the gene differences between the 96 h and 24 h groups. Compared with the control group, there were 1991 significantly up-regulated genes and 1319 significantly down-regulated genes 96 h after irradiation. Compared with 24 h after irradiation, there were 1842 significantly up-regulated genes and 1467 significantly down-regulated genes after 96 h of irradiation. Figure 1C shows the trend of gene expression levels under different time effects, which reveals a significant difference between short- and long-term effects.

We paid special attention to the long-term effects after irradiation and performed Gene Ontology (GO) and Kyoto Encyclopedia of Genes and Genomes (KEGG) enrichment analyses for genes that were significantly up-regulated compared to the control group at 96 h after irradiation. GO enrichment results showed that these up-regulated genes were mainly involved in biological processes and molecular functions related to stress response, immune response, cell growth inhibition, and protein metabolism (Figure 1D). This suggests that irradiation may affect cells on multiple levels, including enhancing immune response, regulating protein synthesis, affecting cell repair and regeneration, and regulating cell cycle and metabolic processes.

KEGG enrichment analysis showed that the up-regulated genes induced by irradiation were involved in several key cell signaling pathways, such as immune response and inflammation regulation, cell stress and apoptosis, cancer-related pathways, metabolic abnormalities, cell death, immune escape, cell senescence, and autophagy. Figure 1E shows the top 15 enrichment pathways and their significance levels (*p*-values). Figure 1F shows the gene enriched to the cellular senescence pathway, with a total of 25 genes enriched in this pathway. These KEGG enrichment results highlight the extensive effects of irradiation on cell function, which not only involves the physiological regulation of cells, but also may affect pathological processes such as cancer, and have a profound impact on long-term tumor control and patient prognosis.

### 3.2. Long-Term Effects of Irradiation Inducing Cellular Senescence

After identifying the short- and long-term gene expression changes induced by medium-dose irradiation, we further investigated the cell response to high-dose irradiation, especially the effects of longer duration on cell biological function. This high-dose irradiation is closer to the conditions used in clinical therapy and can provide clearer evidence to reveal the mechanism of tumor cell tolerance to radiation therapy [20]. A dose of 20 Gy is close to a single high-dose exposure or fractional total dose in some cancer radiation treatments [21]. This dose can simulate the conditions in actual clinical treatment and help study the effects of radiation on tumor cells, thus providing data support for radiotherapy optimization [22,23]. By using high doses of irradiation, it is possible to distinguish more clearly between radio-sensitive and radio-resistant cells.

Mass spectrometry was performed on cell protein samples from the unirradiated (group A), 5 days after 20 Gy irradiation (group B), and 7 days after 20 Gy irradiation (group C) (Appendix A). Figure 2A shows the principal component analysis (PCA) results of these three groups. PCA extracted two main components: the first principal component (PC1) explained 83.66% of the variance, and the second principal component (PC2) explained 9.89% of the variance. The samples of Group A are clustered in the area where PC1 and PC2 are close to the origin, indicating that the characteristics of the samples in this group are similar. In contrast, samples from Group B were clearly distributed in the positive region of PC1, indicating significant differences in major features between them and group A. Group C samples were concentrated on the distal end of PC1 and PC2, which formed a clear distinction from the other two groups.

The bottom of Figure 2A shows the cumulative distribution of the coefficients of variation (cv) for the three sets of data (A, B, and C). The cumulative curves of the three groups overlap almost exactly, indicating that they differ little in terms of variability or data volatility. This highly similar accumulation curve of coefficient of variation indicates that the sample data under different treatments have good stability and consistency, which ensures the credibility of the results of mass spectrometry, and can truly reflect the changes in protein expression under different treatment conditions.

Figure 2B shows the protein expression of the three groups of samples. In the unirradiated group (group A), the expression levels of most proteins were lower; in group B and group C, the expression levels of some proteins changed significantly, the expression of some proteins increased, some proteins decreased, and some proteins maintained relatively stable expression levels. Heat maps show that the effects of irradiation on protein expression are time-dependent. Protein expression was relatively stable under unirradiated conditions, and some protein expression levels gradually increased with the extension of irradiation time, suggesting that the cells may initiate large-scale stress response or repair mechanism under continuous external stimulation. These results provide important clues for further investigation of how irradiation regulates cell biological processes.

Figure 2C visually shows the difference in gene expression at different times after high-dose irradiation. It can be observed that the expression of some genes is similar, but the expression of some genes is different. The difference between group C and group B mainly focused on nuclear proteins (35%), cytoplasmic proteins, and mitochondrial proteins (Figure 2D).

Figure 2E shows the top 10 KEGG enrichment pathways of up-regulated genes in groups B and C compared to group A. Compared with group A, up-regulated gene enrichment in group B was found in multiple pathways, including cell death, apoptosis, cell senescence, cell cycle, immune response, and metabolic pathways. The genes up-regulated in group C compared to group A were concentrated in a variety of biological processes related to cell death, immune response, and metabolic regulation. As can be seen from Figure 2E, necroptosis ranks first in the enrichment pathway. Necroptosis is an acute cell death mechanism in which cells are quickly cleared after death, and while it triggers a strong inflammatory response, it may not have the same impact on the long-term cellular environment as the persistence of senescent cells. Therefore, our research focus turned to cellular senescence.

Cell senescence is when cells stop dividing and enter a stable, non-proliferating state after being severely damaged, such as DNA damage caused by radiation, rather than dying immediately [24]. Senescent cells can exist for a long time and affect the surrounding tissue and cell microenvironment by secreting inflammatory factors and matrix remodeling factors, especially in cancer, senescence-related diseases, and tissue repair [25]. Studying cellular senescence can reveal the effects of radiation on long-term tissue function, including tissue senescence, carcinogenesis, and decreased regenerative capacity.

We compared the expression differences of senescence pathway up-regulated genes in group B and group C. Figure 2F shows the expression of senescence pathway genes at different times after high-dose irradiation. As can be seen from the figure, the expression of some genes in group C was significantly higher than that in group B, indicating that the degree of cell senescence in group C may be more significant. This provides new insights into the long-term effects of high doses of radiation.

### 3.3. ALDH1A3 as a Diagnostic and Prognostic Marker for Prostate Cancer

In radiation therapy studies, exploring the combined effects of different dose and time effects helps to model the mechanism by which tumor cells respond to radiation therapy. By comparing genetic and protein changes at these doses and time points, we can identify radiation-sensitive and resistant cells, as well as potential therapeutic targets and side-effect mechanisms.

We used Venn diagram analysis to identify 110 genes that were affected by long-term effects at different doses (Figure 3A). These 110 genes are enriched into immune-related pathways, cell death and senescence pathways, metabolism and metabolic regulation pathways, etc. Figure 3B shows the top 10 enriched pathways. This suggests that the long-term effects of irradiation involve multiple biological processes such as immune response, cellular senescence, and metabolic regulation, and estimates of these pathways provide important clues to understanding the far-reaching effects of irradiation on cells and tissues.

In order to identify potential cancer markers and study the effects of irradiation on tumors, providing a basis for precise treatment, we then performed an intersection analysis between these 110 genes and the genes highly expressed in the cancer tissues of prostate adenocarcinoma (PRAD) patients in the TCGA database, and identified four genes (Figure 3A, Appendix A). Figure 3C shows the expression of these four genes in cancer and normal tissue of PRAD patients. The results showed that these four genes were highly expressed in cancer tissue and less expressed in normal tissue, suggesting that they may serve as potential cancer markers.

Next, we analyzed the influence of these four genes on the biochemical recurrence-free survival (BRFS) of PRAD patients receiving radiotherapy, using the “surv_cutpoint” function, and divided them into two groups based on the expression of *ALDH1A3* (Figure 3D, Appendix A). We found that patients with high expression of *ALDH1A3* had poor BRFS, suggesting that high expression of *ALDH1A3* may be associated with disease progression or recurrence in patients (Figure 3E).

We further analyzed the expression of *ALDH1A3* in other types of cancer in TCGA and found that the expression of *ALDH1A3* in cancer tissue of PRAD patients was the highest, indicating that the high expression of *ALDH1A3* may play an important role in the occurrence, progression, or malignant transformation of PRAD, and it may be a key regulatory gene in prostate cancer (Figure 3F). Because *ALDH1A3* is expressed higher in prostate cancer than in other cancers, it suggests that it may have some specificity as a diagnostic or prognostic marker for prostate cancer, this specificity helps identify prostate cancer patients, and may be useful for monitoring disease progression or risk of recurrence.

### 3.4. ALDH1A3 Links Age, Immune Response, and Drug Sensitivity in PRAD

By analyzing the expression levels of the *ALDH1A3* gene in PRAD patients of different ages receiving radiotherapy, the results showed that the expression of *ALDH1A3* was higher in patients under 60 years old, while the expression was lower in patients 60 years old and above, with statistically significant differences between the two (Figure 4A). This suggests that the expression of *ALDH1A3* may be regulated by age, thus affecting the response of patients of different ages to radiotherapy and immunotherapy.

To further investigate the effect of *ALDH1A3* expression level on immune cell infiltration, we used a variety of immune cell infiltration evaluation algorithms, including xCell, CIBERSORT, EPIC, and MCPCOUNTER. The xCell algorithm analysis revealed that the infiltration level of CD4^+^ Th1 cells was higher in the group with high expression of *ALDH1A3* (Figure 4B). After analysis using the CIBERSORT algorithm, it was found that the infiltration level of the macrophage M2 subtype was higher in the high-expression group (Figure 4C). The EPIC algorithm showed that the infiltration level of NK cells was higher in the group with low expression of *ALDH1A3* (Figure 4D). The analysis results of the MCPCOUNTER algorithm showed that B cells and monocytes were infiltrated more in the high-expression group, while monocytes were infiltrated more in the low-expression group (Figure 4E,F). In addition, we performed tumor immune dysfunction and exclusion (TIDE) prediction, which showed that the TIDE score was higher in the group with low expression of *ALDH1A3*, suggesting that these patients responded less well to immunotherapy (Figure 4G).

The results of drug sensitivity prediction showed that the high expression group of *ALDH1A3* was sensitive to rapamycin, OSI-027, ribociclib, and azd6482, and the low expression group was sensitive to azd5991 (Figure 4H). These results suggest that the expression level of *ALDH1A3* is not only related to the infiltration of immune cells, but also may affect the sensitivity of patients to different drugs, thus providing a basis for personalized treatment.

### 3.5. Radiation Induces Cellular Senescence and Triggers SASP by Activating the cGAS–STING Pathway

The above studies have shown that the expression level of *ALDH1A3* is closely related to immune cell infiltration and drug sensitivity, especially patients with low expression of *ALDH1A3* show poor immunotherapy response, and the cells in the low expression group may be closely related to cellular senescence. Senescent cells are often associated with immune escape and chronic inflammation, which is consistent with the observation of high TIDE scores and poor response to immunotherapy.

To better understand the long-term effects of radiation on prostate cancer cells, we further explored the effects of irradiation on cellular senescence. In the long-term effects of irradiation, we found that many up-regulated genes were enriched in the cellular senescence pathway. To verify this phenomenon, the cells were examined using SA-β-gal staining and the results showed that the staining of the cells gradually deepened over time, indicating a significant increase in senescence (Figure 5A). In addition, the expression of senescence marker genes *CDKN2A* (*p16*) and *CDKN1A* (*p21*) also gradually increased over time, further supporting the irradiation-induced increased senescence (Figure 5B).

During senescence, cells often maintain chronic inflammatory states through the cGAS–STING pathway and affect the surrounding environment through the secretion of SASP [13]. A key feature of cellular senescence is the decline in the integrity of the nuclear membrane, especially the functional damage or dysregulation of nuclear membrane proteins [26]. This leads to increased vulnerability of the nuclear membrane and the escape of DNA fragments into the cytoplasm, a phenomenon common during senescence and closely associated with DNA damage responses. cGAS is a DNA sensor inside cells. When cells are senescent or damaged, DNA in the nucleus will leak into the cytoplasm, and cGAS can recognize and bind these DNA fragments, thereby activating the STING signaling pathway [15,27]. Activated STING can initiate a series of immune responses, induce cells to produce interferon and pro-inflammatory factors, and maintain the chronic inflammatory state of cells [27].

cGAS–STING pathway maintains SASP secretion and a persistent inflammatory state by activating type I interferon and pro-inflammatory factors [15]. This process plays a key role in the development of inflammatory diseases and cancer progression. In some cases, overactivation of the cGAS–STING pathway can keep the immune system at a low level of activation for a long time, prompting senescent cells to escape immune surveillance and even accelerate tumorigenesis [28]. In addition, the secretion of SASP may promote tumor growth and spread by altering the tumor microenvironment [29].

cGAS–STING pathway induces an immune response by activating type I interferon, and type I interferon activates STAT2 through the JAK–STAT signaling pathway [30]. The activated STAT2 forms a complex with STAT1 and IRF9 to regulate the expression of interferon-stimulated genes and further enhance immune and inflammatory responses [30]. This amplifying effect drives the production of SASP, exacerbating chronic inflammation.

Studies have shown that the cGAS–STING pathway plays a crucial role in maintaining chronic inflammation and regulating cellular senescence [28]. We monitored key proteins in this pathway through Western blot, and the results showed that cGAS, p-STING, TBK1, p-TBK1, IRF3, STAT2, p65, and other proteins were significantly up-regulated in senescent cells. This is consistent with other studies showing that the cGAS–STING pathway is widely active in senescent cells, further confirming the importance of this pathway in regulating cellular inflammatory response and cancer progression [31,32,33,34]. (Figure 5C,D). These results suggest that cGAS senses DNA in the cytoplasm and activates STING, which in turn promotes the production of type I interferon and inflammatory response through downstream signaling molecules such as TBK1 and IRF3. Activation of STAT2 suggests that the interferon signaling pathway is also activated, while up-regulation of p65 suggests that the NF-κB pathway is involved in the expression of pro-inflammatory factors. This is consistent with the characteristics of SASP secretion and chronic inflammation in senescent cells, revealing the important role of the cGAS–STING pathway in senescence-related inflammation and diseases.

In addition, we observed that type I interferons eventually induced up-regulation of MHC I expression in several different prostate cancer cell lines induced by irradiation (Figure 5E), suggesting that type I interferons play a key role in immune regulation after irradiation. This finding further suggests that type I interferons not only play a regulatory role in the senescence process, but also play an important role in immune responses in the tumor microenvironment.

### 3.6. ALDH1A3 Depletion Accelerates Cellular Senescence

The above studies revealed the key role of the cGAS–STING pathway in senescent cells, especially through the maintenance of SASP secretion and chronic inflammatory state, affecting tumor microenvironment and cancer progression. Considering that *ALDH1A3* may play a role in the regulation of cGAS–STING pathway activation and SASP secretion, we speculated that its expression level has a direct impact on the inflammatory response of senescent cells. First, we conducted Spearman’s rho analysis on publicly available prostate cancer clinical samples to evaluate the potential correlation between *ALDH1A3* and SASP genes; however, the results did not reveal a significant association (Appendix A).

Therefore, in order to further investigate the mechanism of action of *ALDH1A3* in cell senescence, especially its regulation of the cGAS–STING pathway before and after radiation, we designed a series of experiments to verify its function. First, we used CRISPR-Cas9 technology to knock down the *ALDH1A3* gene, and the knock-down effect is shown in Figure 6A. Next, cells transfected with sgRNA were subjected to either unirradiation or irradiation treatment, and SA-β-gal staining was performed to detect the senescence phenotype of the cells (Figure 6B). In the unirradiation group, we found that the staining was significantly deepened in cells that knocked down *ALDH1A3*, indicating that the knocking down of *ALDH1A3* has led to the occurrence of cell senescence even in the absence of irradiation (Figure 6C, left).

In the irradiation group, we observed increased staining intensity in both control and knockdown cells, indicating that irradiation induced a distinct senescence phenotype in both groups. However, statistical analysis showed no significant difference in staining depth between the knockdown group and the control group after irradiation (Figure 6C, right). This indicates that the knockdown of *ALDH1A3* did not further aggravate the senescence phenotype under irradiation.

To further verify the regulatory role of *ALDH1A3* in the cGAS–STING pathway, Western blot analysis was performed on unirradiated cells. The results showed that after knocking down *ALDH1A3*, the expression levels of key proteins in the CGAS–STING pathway, such as cGAS, STING, and TBK1, were significantly lower than those in the control group. This suggests that *ALDH1A3* may play an important role in regulating the activation of the cGAS–STING pathway (Figure 6D). By inhibiting the activation of this pathway, *ALDH1A3* knock-down may reduce inflammatory cytokines released by senescent cells.

These results further support the hypothesis that *ALDH1A3* influences SASP secretion by regulating the cGAS–STING pathway. Although *ALDH1A3* knockdown promoted cell senescence, inflammation, and SASP secretion did not intensify and even decreased due to the inhibition of the cGAS–STING pathway. This suggests that *ALDH1A3* not only plays a role in regulating cellular senescence but also participates in the regulation of SASP by influencing the cGAS–STING pathway, which may have a significant impact on the tumor microenvironment.

### 3.7. Depletion of ALDH1A3 Inhibits SASP-Associated Tumor Growth

Although senescent cells themselves do not proliferate, they are theoretically expected to prevent carcinogenesis. However, the SASP factors secreted by senescent cells may promote the growth and spread of cancer cells. Previous experimental results have shown that while the knockout of *ALDH1A3* can promote cellular senescence, its inhibition of the cGAS–STING pathway may reduce SASP secretion. Therefore, we investigated whether the senescent cells induced by the knockdown of *ALDH1A3* could promote the in vitro proliferation and migration of cancers. By assessing the role of SASP in these functions, we can further elucidate whether *ALDH1A3* plays a broader role in cellular senescence through the regulation of SASP.

To verify the impact of *ALDH1A3* knockout on the SASP function of senescent cells, we designed and conducted relevant experiments. We used a transwell co-culture system to physically separate senescent cells and cancer cells, while allowing cancer cells to receive the SASP secreted by senescent cells. Figure 7A shows the detailed operational flow of the experiment. First, we used CCK-8 to assess the effects of control and *ALDH1A3* knockout on cell proliferation at 0, 24, 48, and 72 h. As expected, the *ALDH1A3* knockout group did not significantly promote cell proliferation, and there were no significant differences at any time point compared to the control group (Figure 7B,C). Statistical analysis of the data after 72 h of co-culture revealed no significant differences in cell proliferation capability between the knockout and control groups (Figure 7D). Specifically, the *p*-values for the comparisons between the sg*ALDH1A3*-1 and sgCTL, sg*ALDH1A3*-2 and sgCTL, and sg*ALDH1A3*-3 and sgCTL groups were 0.3, 0.33, and 0.42, respectively. These results indicate that the senescent cells induced by *ALDH1A3* knockout had a limited promoting effect on cancer cell proliferation.

Next, we evaluated the effect of the SASP components secreted by senescent cells induced by *ALDH1A3* knockdown on cancer cell migration using a scratch assay (Figure 7E). After 24 h of co-culture, we performed statistical analysis on the healing area. The results showed that there were no statistically significant differences between the *ALDH1A3* knockout group and the control group in promoting cancer cell migration (Figure 7F). Specifically, the *p*-values for the comparisons between the sg*ALDH1A3*-1 and sgCTL, sg*ALDH1A3*-2 and sgCTL, and sg*ALDH1A3*-3 and sgCTL groups were 0.45, 0.38, and 0.65, respectively. This indicates that *ALDH1A3* knockout did not significantly alter the influence of SASP on cancer cell migration.

Overall, these results suggest that senescent cells knocked down by *ALDH1A3* inhibit the promoting effect of SASP on cancer progression. This result may be due to impaired cGAS–STING pathways. Therefore, *ALDH1A3* plays an important role in controlling the harmful behavior of senescent cells, and targeting *ALDH1A3* may be a new strategy to combat the diseases induced by SASP.

## 4. Discussion

In this study, we investigate the regulatory role of *ALDH1A3* in radiotherapy for prostate cancer, especially its effects on cellular senescence and SASP. Our findings demonstrate that *ALDH1A3* gene knockout accelerated the cellular senescent-like phenotype, but inhibits SASP’s promoting effect on cancer by suppressing the cGAS–STING immune pathway. This finding suggests that although *ALDH1A3* knockout promotes cell senescence, tumor progression can be slowed by regulating SASP. The results reveal the molecular mechanism of *ALDH1A3* in prostate cancer cells. This suggests that *ALDH1A3* may be a potential therapeutic target to improve radiotherapy efficacy and control the adverse effects of SASP.

As a common form of cancer treatment, radiation therapy mainly uses high-energy rays to damage the DNA of cancer cells, thereby inhibiting their growth and division [35,36]. Radiation therapy can be used to treat many types of cancer, either as the primary treatment or in combination with surgery or chemotherapy to improve treatment effectiveness. Although the main goal of radiotherapy is to directly kill cancer cells, its mechanism of action is far more complex than simple DNA damage, especially in the field of cell senescence research has attracted increasing attention [37,38,39].

In recent years, the phenomenon of cancer cell senescence caused by radiotherapy has become an important research field. Radiotherapy uses high-energy radiation to damage the DNA of cancer cells, forcing them to activate repair mechanisms [35,40]. However, when DNA damage is too severe to be fully repaired, cells may enter a state called senescence. This is an irreversible cell cycle arrest phenomenon, that is, although the cell loses the ability to proliferate, it still maintains metabolic activity and continues to persist in the body [41,42].

Radiation-induced cellular senescence typically involves several key signaling pathways, including immune response regulation, inflammation, cell stress and apoptosis, cancer-related pathways, metabolic abnormalities, cell death, immune escape, and autophagy. Among them, the p53/p21, p16/Rb, and other pathways play a crucial role in cell cycle regulation and the DNA damage response [43,44,45]. The changes in cell metabolism, apoptosis, and immune response can affect the tumor microenvironment, influencing tumor development and therapeutic efficacy. When cancer cells undergo necrotic apoptosis induced by various treatments, neutrophils and macrophages in the tumor microenvironment are stimulated to produce immune suppression through the release of interleukin-1α (IL-1α), which partially explains the limited effectiveness of combined radiotherapy, chemotherapy, and immunotherapy [46].

In addition, radiation-induced cell senescence is associated with the appearance of an SASP. SASP means that senescent cells secrete a range of inflammatory factors, chemokines, and proteases that not only alter the tumor microenvironment, but may also affect neighboring healthy cells and even, in some cases, promote cancer cell survival and metastasis [9]. Various forms of metabolic stress can both promote aging and influence the SASP. The senescent cytoplasm contains high levels of polyunsaturated fatty acids, which can contribute to the lipid-based SASP [47].

Although radiation-induced cell senescence can effectively inhibit the proliferation of cancer cells in the short term, its long-term effects have aroused the attention of researchers. Some studies suggest that SASP may increase the risk of cancer recurrence by proinflammatory and remodeling the microenvironment, and may even lead to new cancers [29,48,49]. Therefore, how to induce cell senescence while avoiding the negative effects of SASP has become an important research topic. Here, we find that *ALDH1A3* plays a key role in cellular senescence as well as the production of SASP, and describe the molecular mechanism by which *ALDH1A3* mediates this effect.

We investigated the changes in gene and protein expression of cells after different doses of irradiation. Specifically, we focused on the short- and long-term effects of medium (5 Gy) and high (20 Gy) doses of irradiation. We further studied the role of *ALDH1A3* and its relationship with cell senescence, SASP secretion, and cGAS–STING pathway. These studies provide new perspectives for understanding the regulation of cell biological functions by irradiation and reveal the potential role of *ALDH1A3* in cell senescence and immune regulation.

First, our results show that irradiation has an effect on the gene expression patterns of cells, and that the effects differ at different time effects. Studies of high-dose irradiation further revealed the time-dependent effect of cellular protein expression. Mass spectrometry showed significant differences in protein expression at different time points after irradiation. Notably, there were marked changes in nuclear, cytoplasmic, and mitochondrial proteins. The changes in the expression of these proteins indicate that the cells initiate different stress responses and repair mechanisms under sustained irradiation pressure. Enrichment analysis showed that the up-regulated genes after high-dose irradiation were closely related to necrotic apoptosis and cell senescence pathways. Notably, group C (7 days after irradiation) cells exhibited a more significant senescence phenotype. These results suggest that high-dose irradiation causes acute cell death. It may also affect the long-term physiological state of cells and the tumor microenvironment by inducing cell senescence.

In our study of *ALDH1A3*, we found that its high expression in prostate cancer cells is closely related to the cell’s radiation tolerance and disease prognosis. Patients with high expression of *ALDH1A3* showed poor biochemical relapse-free survival, suggesting its potential as a diagnostic and prognostic marker. Analysis of *ALDH1A3* in relation to age and immune cell infiltration revealed its differential expression in patients of different ages. This may have a potential impact on the response to immunotherapy. Specifically, high expression of *ALDH1A3* was associated with high infiltration of CD4^+^ Th1 cells and macrophage M2, while patients with low expression showed stronger infiltration of NK cells. This suggests that *ALDH1A3* may influence patients’ response to radiotherapy or immunotherapy by regulating immune cells in the tumor microenvironment. However, it is important to note that the immune infiltration analysis in this study is primarily based on bioinformatics predictions using computational algorithms such as xCell, CIBERSORT, EPIC, and MCPCOUNTER. While these methods provide valuable insights into immune cell composition, further experimental validation is needed in the future. Therefore, future studies should incorporate experimental techniques such as flow cytometry, immunohistochemistry (IHC), or single-cell sequencing (scRNA-seq) to confirm the predicted immune cell infiltration patterns and further assess their biological relevance in prostate cancer.

More importantly, our study shows that *ALDH1A3* affects cell senescence and SASP secretion by regulating the cGAS–STING pathway. The key role of the cGAS–STING pathway in maintaining chronic inflammation and cellular senescence has been confirmed by multiple studies, and our results further support this view. The *ALDH1A3* knock-down experiment showed that knocking down *ALDH1A3* accelerated cell senescent-like phenotype. However, it also inhibited the activation of the cGAS–STING pathway, reducing the secretion of pro-inflammatory factors in SASP. This finding suggests that *ALDH1A3* not only affects cell division arrest during the regulation of cell senescence, but also regulates inflammatory response through the cGAS–STING pathway, which may play a more extensive role in tumor microenvironment and senescence-related diseases.

Although our study reveals the important role of *ALDH1A3* in cellular senescence and radioresistance, there are still some questions to be further investigated. First, we demonstrate that *ALDH1A3* knockdown affects SASP secretion. However, different regulatory mechanisms may exist for SASP components in different cell types or cancer types. Secondly, how *ALDH1A3* affects other key cancer-related pathways through the cGAS–STING pathway remains to be further explored. In addition, the targeted therapy of *ALDH1A3* for tumors needs to be evaluated in vivo and in vitro for its toxic side effects after inhibition. Furthermore, the potential for combining *ALDH1A3* inhibition with existing immune checkpoint inhibition therapy or senescent cell purgants to improve tumor patient prognosis needs to be verified through further cell, animal, and clinical studies.

## 5. Conclusions

In conclusion, this study highlights the dual role of *ALDH1A3* in prostate cancer cells, where it regulates both cellular senescence and the SASP. *ALDH1A3* not only plays a role in maintaining radiotherapy resistance but also affects tumor progression by modulating the inflammatory responses associated with senescence. Targeting *ALDH1A3* may offer a new therapeutic strategy to enhance radiotherapy efficacy while controlling the deleterious effects of SASP, making it a promising avenue for future cancer treatments.

## Figures and Tables

**Figure 1 cancers-17-01184-f001:**
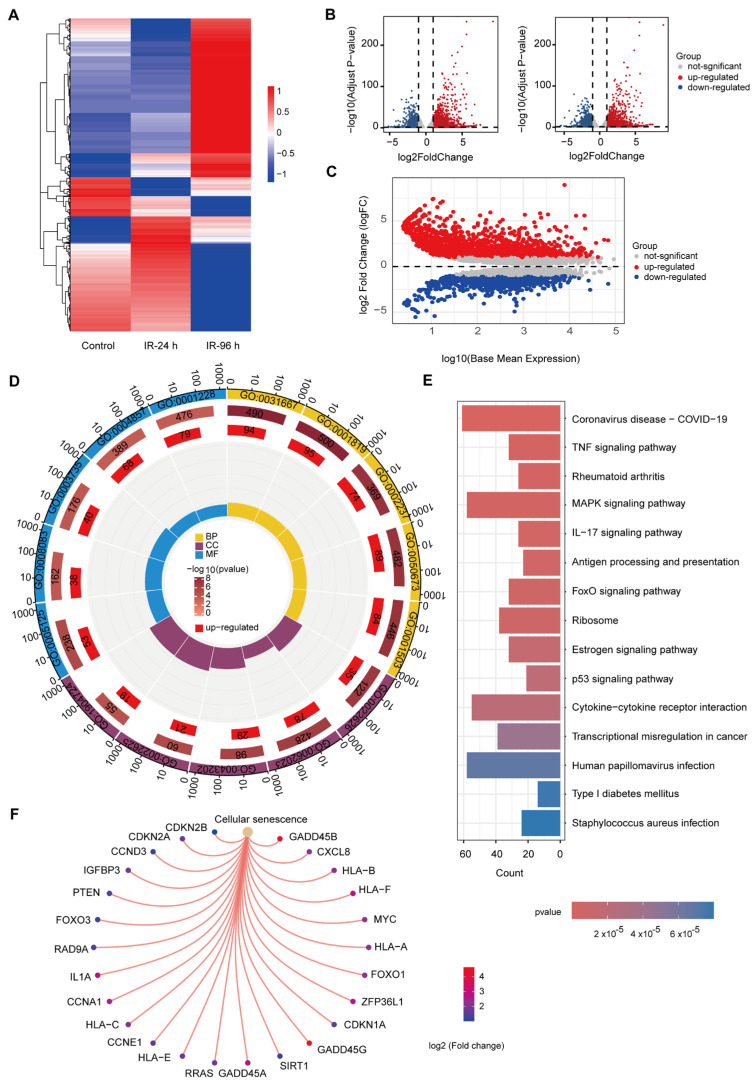
Comprehensive analysis of gene expression changes in PC3 cells at short- and long-term intervals after irradiation. (**A**) Genes from unirradiated PC3 cells (Control), 24 h (IR-24 h), and 96 h (IR-96 h) post-irradiation were clustered using RNA sequencing data. (**B**) Volcano plot illustrating gene expression changes from RNA sequencing. The left side shows differential expression in IR-96 h vs. Control, while the right side highlights changes in IR-96 h vs. IR-24 h. (**C**) MA plot showing differential gene expression between IR-96 h and IR-24 h in PC3 cells. (**D**) GO enrichment analysis of up-regulated genes in IR-96 h. (**E**) Top 15 KEGG pathways enriched by up-regulated genes in IR-96 h. (**F**) Enrichment of IR-96 h up-regulated genes in the cellular senescence pathway.

**Figure 2 cancers-17-01184-f002:**
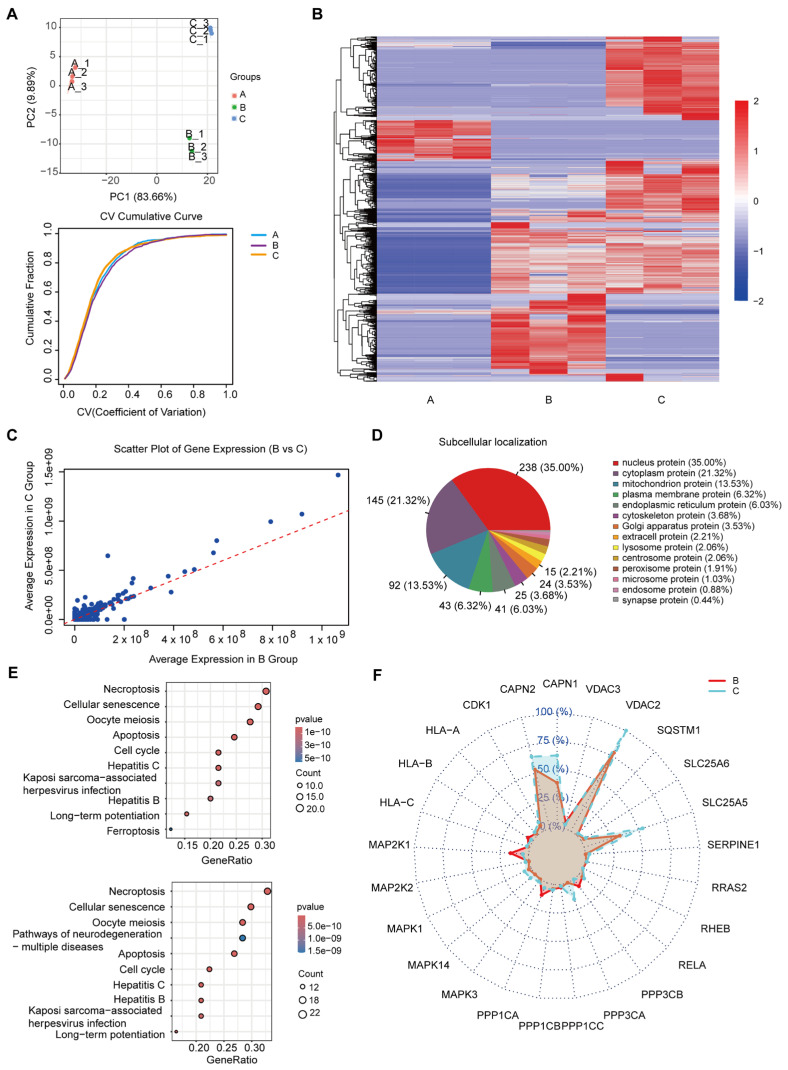
Integrated analysis of protein expression, pathway enrichment, and cellular senescence in irradiated groups. (**A**) PCA and CV analysis of protein expression across irradiation groups. Top: PCA of protein expression across three groups: unirradiated (Group A), 5 days post-irradiation with 20 Gy (Group B), and 7 days post-irradiation with 20 Gy (Group C). Bottom: results of CV analysis for these three groups. (**B**) Clustering heatmap of protein expression profiles for the three groups. (**C**) Scatter plots of protein expression levels in Group B and Group C. (**D**) Distribution of different proteins between Group B and Group C. (**E**) KEGG enrichment analysis of up-regulated proteins in Groups B and C compared to Group A. Top: the top 10 pathways of up-regulated gene enrichment in Group B compared to Group A. Bottom: the top 10 pathways of up-regulated gene enrichment in Group C compared to Group A. (**F**) Expression of key genes in the cellular senescence pathway.

**Figure 3 cancers-17-01184-f003:**
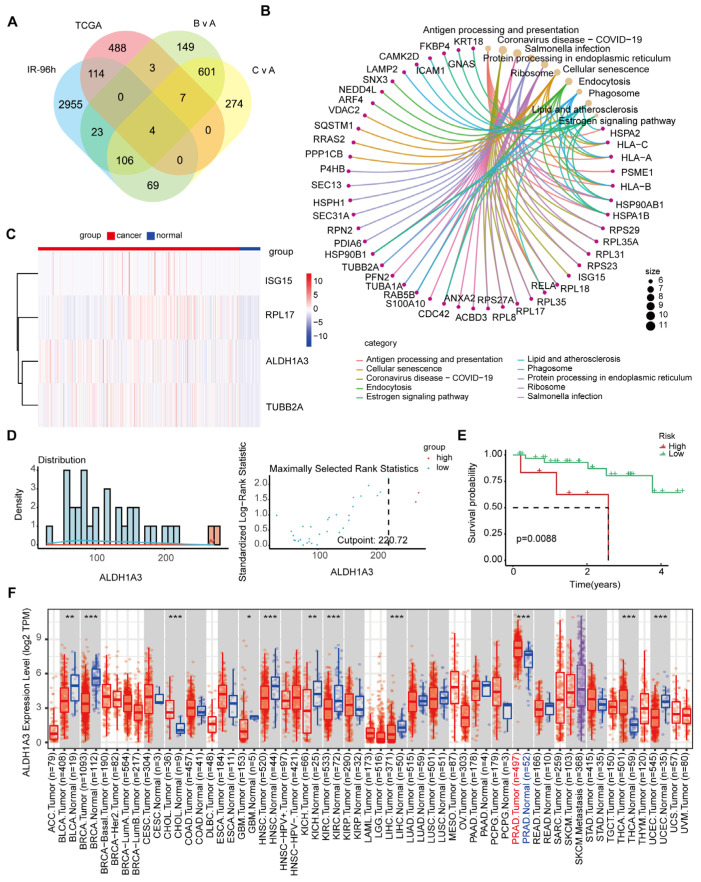
Identification of radiation-affected genes and their prognostic significance in prostate cancer. (**A**) Venn diagram showing the intersection of up-regulated genes from long-term irradiation exposure at different doses (5 Gy and 20 Gy) with up-regulated genes in prostate cancer tissue from the TCGA database. (**B**) Top 10 KEGG enrichment pathways of the 110 genes up-regulated in irradiated cells. (**C**) Expression analysis of four key genes identified from the intersection of irradiation-induced up-regulated genes and highly expressed genes in prostate cancer. (**D**) Determination of the optimal cutoff value for *ALDH1A3* expression using the “surv_cutpoint” function. (**E**) Kaplan-Meier BRFS curves for high and low expression groups of *ALDH1A3*. (**F**) Analysis of *ALDH1A3* expression across different cancer types in the TCGA database. *** *p* < 0.001; ** *p* < 0.01; * *p* < 0.05.

**Figure 4 cancers-17-01184-f004:**
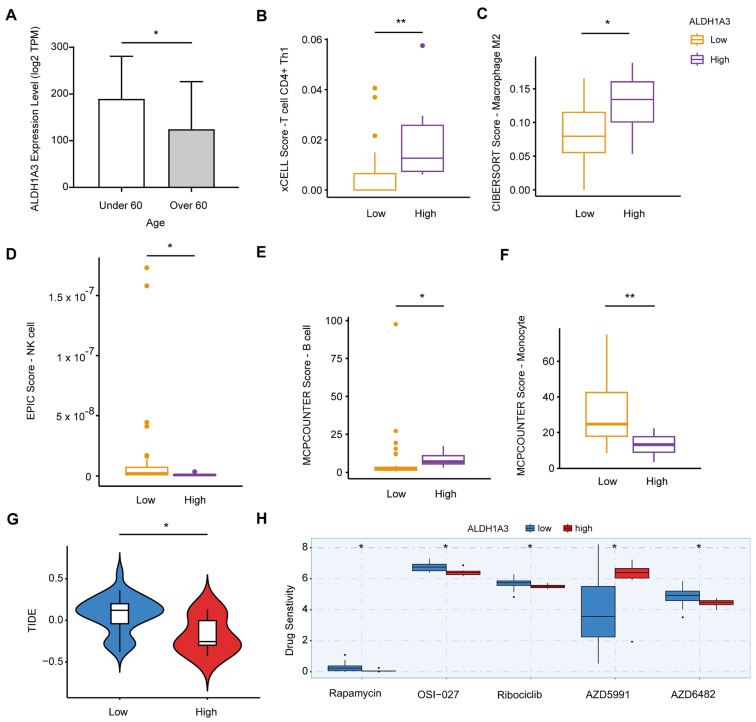
Impact of *ALDH1A3* expression on immune infiltration and drug sensitivity. (**A**) Comparison of *ALDH1A3* expression levels in PRAD patients under and over the age of 60. n = 39 patients (under 60, n = 12; over 60, n = 27). *p* = 0.0140, effect Size: r = 0.39 (medium effect size). (**B**) Differences in xCell scores of T cell CD4+ Th1 cells between low and high *ALDH1A3* expression. Cohen’s d = −1.30 (large effect size). (**C**) Differences in CIBERSORT scores of macrophage M2 cells between low and high *ALDH1A3* expression. Cohen’s d = −1.08 (large effect size). (**D**) Differences in EPIC scores of NK cells between low and high *ALDH1A3* expression. Cohen’s d = 0.3988 (medium effect size). (**E**,**F**) Differences in MCPCOUNTER scores of B cells and monocytes between low and high *ALDH1A3* expression. Cohen’s d = −0.09 (small effect size). (**F**) Differences in MCPCOUNTER scores of monocytes between low and high *ALDH1A3* expression. Cohen’s d = 1.11 (large effect size). (**G**) TIDE analysis predicted response to immunotherapy in patients with high or low expression of *ALDH1A3*. Cohen’s d = 1.14 (large effect size). (**H**) Prediction of drug sensitivity in patients with high or low expression of *ALDH1A3*. n = 39 patients (high expression, n = 6; low expression, n = 33). ** *p* < 0.01; * *p* < 0.05.

**Figure 5 cancers-17-01184-f005:**
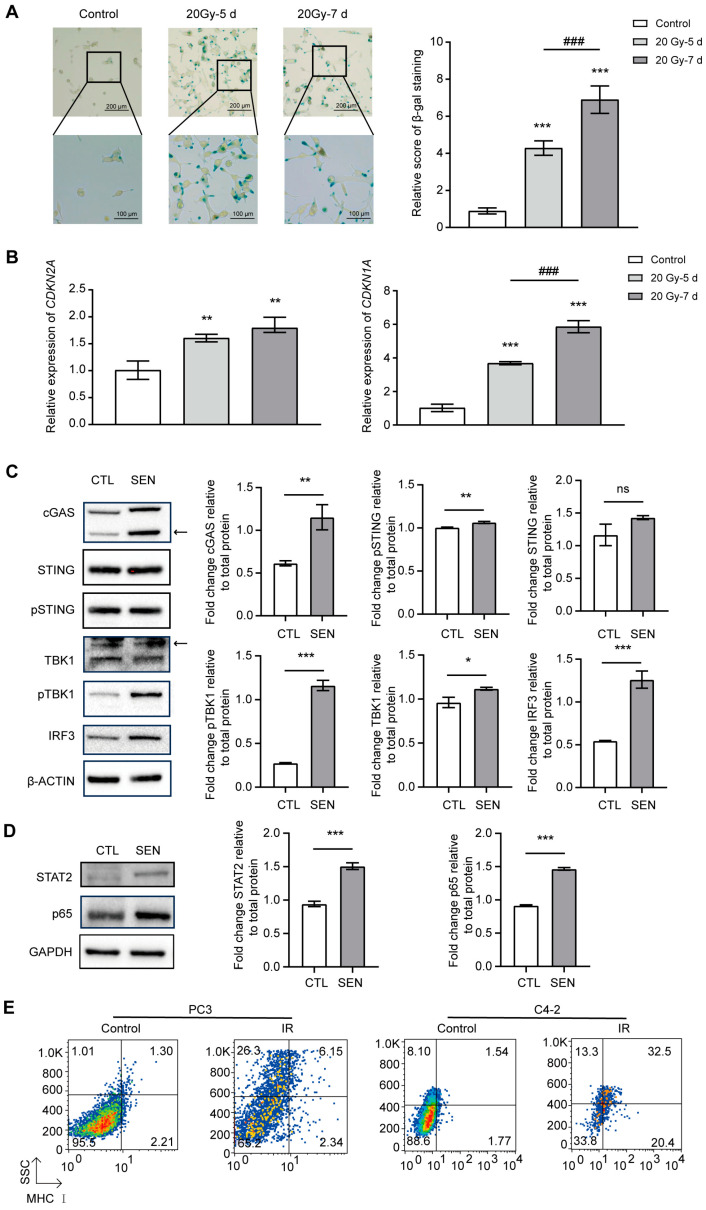
Irradiation-induced cellular senescence and activation of the cGAS–STING pathway. (**A**) SA-β-gal staining results. (**left**) SA-β-gal was assessed using X-gal, with non-specific bands indicated by arrows. (**right**) Quantification of SA-β-gal staining. (**B**) RT-qPCR analysis of CDKN2A and CDKN1A expression. Each sample was repeated independently three times and normalized to GAPDH. (**C**,**D**) Western blot analysis of key proteins in the cGAS–STING pathway. (**left**) Cell lysates were subjected to Western blot against indicated proteins. Arrows indicate non-specific bands. (**right**) Quantification of relative protein levels. (**E**) MHC I expression analyzed by flow cytometry. Data are presented as mean  ±  SD from three independent experiments. *** *p* < 0.001, ** *p* < 0.01, * *p* < 0.05, ### *p* < 0.001, ns (non-significant). The uncropped blots are shown in Appendix A.

**Figure 6 cancers-17-01184-f006:**
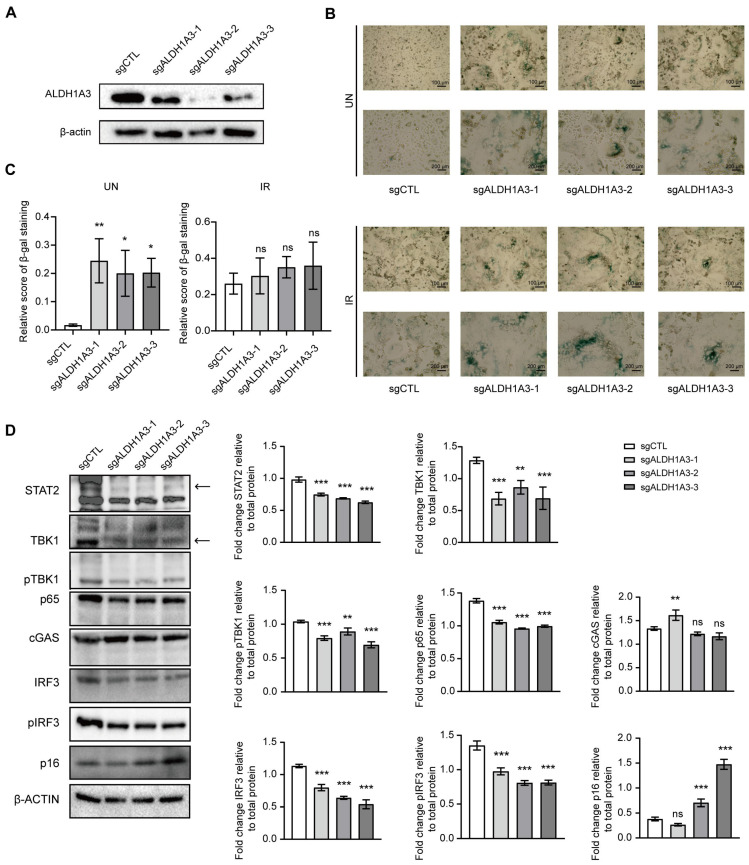
*ALDH1A3* knockdown effects on cellular senescence and the cGAS–STING pathway. (**A**) Western blot validation of *ALDH1A3* knockdown. PC3 cells were treated with sgRNA: control group (sgCTL) and knockdown group (sg*ALDH1A3*). (**B**) SA-β-gal staining of unirradiated (UN) and irradiated (IR) control cells and *ALDH1A3* knockdown cells. (**C**) Quantitative analysis of SA-β-gal staining intensity in UN and IR cells. (**D**) Western blot analysis of key proteins in the cGAS–STING pathway. (**left**) Cell lysates were subjected to Western blot analysis against the indicated proteins. Arrows indicate the bands of target proteins. (**right**) Quantification of relative protein levels. Data are presented as mean  ±  SD of three independent experiments. *** *p* < 0.001, ** *p* < 0.01, * *p* < 0.05, ns (non-significant). The uncropped blots are shown in Appendix A.

**Figure 7 cancers-17-01184-f007:**
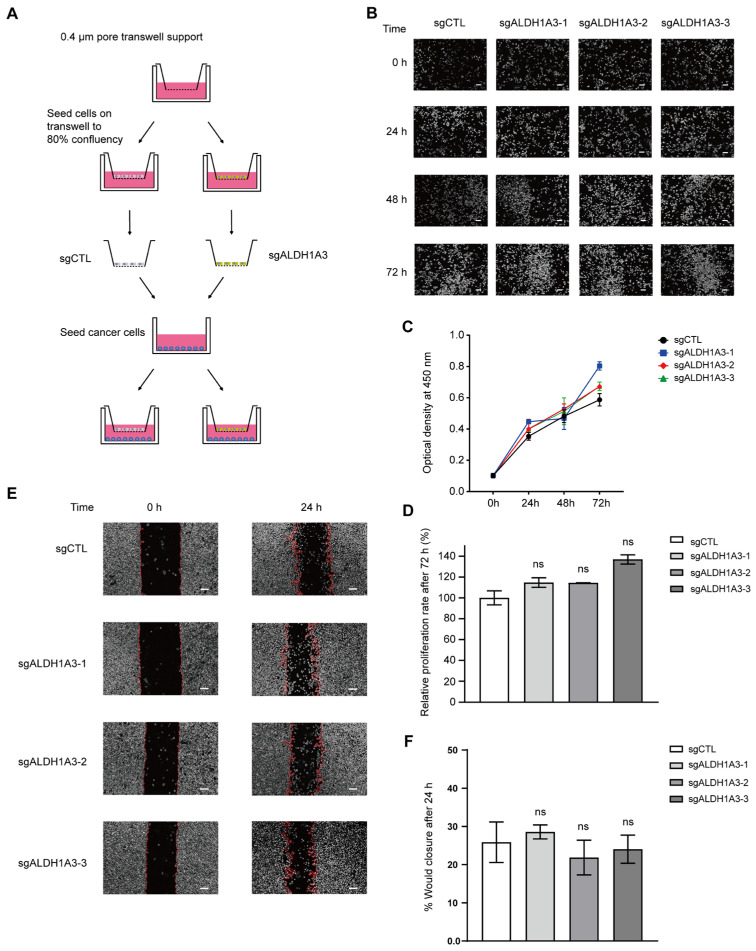
*ALDH1A3*-depleted senescent cells do not promote senescence-associated tumor growth in vitro. (**A**) Schematic diagram of the transwell co-culture setup. (**B**) Co-culture images of PC3 cells with either non-senescent cells treated with control (sgCTL) or senescent cells with *ALDH1A3* knockdown (sg*ALDH1A3*). Scale bar, 200 μm. (**C**) CCK-8 assay results showing the proliferation of PC3 cells co-cultured with sgCTL- or sg*ALDH1A3*-treated cells at 0, 24, 48, and 72 h. (**D**) Statistical analysis of cell proliferation after 72 h of co-culture. sg*ALDH1A3*-1 vs. sgCTL, *p* = 0.1262, sg*ALDH1A3*-2 vs. sgCTL, *p* = 0.0945, sg*ALDH1A3*-3 vs. sgCTL, *p* = 0.3333. (**E**) Images of PC3 cells immediately after scratching (0 h) and 24 h post-scratching (24 h), co-cultured with either sgCTL- or sg*ALDH1A3*-treated cells. Scale bar, 200 μm. (**F**) Percentage of wound healing after 24 h of co-culture with PC3 cells and cells treated with sgCTL or sg*ALDH1A3*. sg*ALDH1A3*-1 vs. sgCTL, *p* = 0.4528, (c) sg*ALDH1A3*-2 vs. sgCTL, *p* = 0.3784, (d) sg*ALDH1A3*-3 vs. sgCTL, *p* = 0.6511. Data are presented as mean  ±  SD of three independent experiments. ns (non-significant).

## Data Availability

All data supporting this study are provided in the published article and its Appendix A. Permission from the corresponding author is required for data usage.

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
