# Peer review of "ALDH1A3 Regulates Cellular Senescence and Senescence-Associated Secretome in Prostate Cancer"

_cancers, 2025, doi:10.3390/cancers17071184_

Round 1
Reviewer 1 Report
Comments and Suggestions for Authors
The manuscript from S. Wang and Y. Zhao entitle "ALDH1A3 regulates cellular senescence and senescence-associated secretome in prostate cancer" presents a novel investigation into the role of ALDH1A3 in regulating cellular senescence and the senescence-associated secretory phenotype (SASP) in prostate cancer, particularly in the context of radiotherapy. The study shows that ALDH1A3 knockdown accelerates cellular senescence while mitigating SASP-associated tumor progression via inhibition of the cGAS-STING pathway. While majority the findings are well-supported by experiments, the manuscript could benefit from additional depth analysis, particularly regarding senescence markers and downstream signaling events beyond SASP secretion. The study’s clinical relevance is promising but the translational potential of targeting ALDH1A3 could be further elaborated. The statistical analysis and figure presentation need improvement, especially the inclusion of clearer annotations, better figure presentation, and justification for the statistical tests used. Overall, the study makes a good contribution to understanding the interplay between radiation, senescence, SASP, and cancer progression, but addressing these points would enhance the manuscript's scientific rigor and impact.
Major comments:
Introduction: The introduction provides a well-structured overview of the relationship between radiotherapy, senescence, and SASP, highlighting their relevance in cancer progression and therapy resistance. However, it would benefit from addressing the challenges in targeting senescent cells clinically. Specifically, the authors should discuss the dual role of senescence: while it suppresses tumor growth by halting cell proliferation, the SASP can promote chronic inflammation and tumor progression. Challenges in identifying beneficial versus detrimental senescent cells using specific markers should be discussed. The lack of selective senescence markers and concerns about the off-target effects of senolytics should also be included to provide a balanced context for the study.
Results:
- Figure 1: The KEGG pathway analysis identifies enrichment in necroptosis, immune response, and metabolic pathways. However, the authors should discuss the biological significance of these pathways in prostate cancer progression and senescence. For instance, how does necroptosis influence senescence and tumor microenvironment remodeling? Are there specific pathway genes that warrant further investigation?
- Figure 3F: While the claim that ALDH1A3 is a diagnostic marker is intriguing, it relies on expression data from a single cohort. To strengthen this claim, can the authors validate their findings using independent cohorts, such as patient-derived samples or publicly available datasets like GEO. Additionally, the authors could use ROC (Receiver Operating Characteristic) curve analysis to evaluate the sensitivity and specificity of ALDH1A3 as a marker.
- Beta-Galactosidase (SA-β-gal) staining is a widely used marker for cellular senescence, but it is not specific to senescence and can also indicate lysosomal activity in non-senescent cells. The authors’ conclusion of "accelerated senescence" solely based on SA-β-gal staining is not robust. The study should either include additional markers of senescence (e.g., p16, p21, IL6, IL8, IL1α, CXCL1, MMP9) to confirm the phenotype or reframe the findings as indicative of a "senescent-like phenotype." This adjustment is critical to ensure scientific accuracy.
- Figure Format: Several figures (e.g., Figures 3 and 6) appear to be inverted in the manuscript. This reduces the clarity of the data presentation. The authors should adjust the content and ensure that each figure is correctly displayed.
- Clarify and Enhance Figure Annotations: Many figures lack clear annotations, such as statistical significance levels or the rationale behind the specific comparisons made. Each figure should clearly highlight key findings for improved readability.
- For Figure 4: Strengthen statistical analyses by Including sample sizes (n) in figure legends. Reporting effect sizes to complement p-values.
- Clinical Implications: Expand on the therapeutic potential of targeting ALDH1A3. For instance, what are the challenges or risks of ALDH1A3 inhibition? Could this strategy be combined with existing treatments, such as senolytics or immune checkpoint inhibitors, to enhance efficacy? Are there any study limitation that the author can foresee?
The quality of English language in the manuscript is generally clear and professional, making the content understandable. However, there are occasional grammatical errors, awkward phrasing, and areas where the flow could be improved. Some sentences are long and complex, particularly in the discussion and results sections, which could benefit from being broken into shorter, more concise sentences. Certain phrases could also be refined for precision, such as replacing "dye deepening" with "increased staining intensity" to align with standard scientific terminology. Additionally, figure legends sometimes use informal or vague terms, which should be revised.
Author Response
Comments 1:
Introduction: The introduction provides a well-structured overview of the relationship between radiotherapy, senescence, and SASP, highlighting their relevance in cancer progression and therapy resistance. However, it would benefit from addressing the challenges in targeting senescent cells clinically. Specifically, the authors should discuss the dual role of senescence: while it suppresses tumor growth by halting cell proliferation, the SASP can promote chronic inflammation and tumor progression. Challenges in identifying beneficial versus detrimental senescent cells using specific markers should be discussed. The lack of selective senescence markers and concerns about the off-target effects of senolytics should also be included to provide a balanced context for the study.
Response 1: Thank you for your valuable suggestions. Your comments have significantly helped us enhance the content of the article. In response to your suggestion, we have revised the second paragraph of the introduction to include a discussion on the dual role of aging and the associated aging markers. We believe these additions improve the logical flow of the paper and provide a clearer understanding of the critical role of aging following radiotherapy. Once again, thank you for your thoughtful and thorough feedback.
Comments 2:
Figure 1: The KEGG pathway analysis identifies enrichment in necroptosis, immune response, and metabolic pathways. However, the authors should discuss the biological significance of these pathways in prostate cancer progression and senescence. For instance, how does necroptosis influence senescence and tumor microenvironment remodeling? Are there specific pathway genes that warrant further investigation?
Response 2: Thank you for your professional suggestions on our manuscript. We have added the relevant content in the fourth paragraph of the discussion section. Regarding your second question, the KEGG pathways enriched in Figure 1E are primarily derived from genes upregulated in our RNA-seq analysis. These genes serve as candidates for our further research. However, due to the large number of candidates, we also incorporated genes identified through proteomics as upregulated in prostate cancer cells post-radiotherapy, as well as genes that are upregulated in prostate cancer tissue compared to normal tissue in the TCGA database. After this, we narrowed down the list to four candidate genes. Survival analysis of these four genes revealed that only ALDH1A3 was associated with patient prognosis. Therefore, we have selected ALDH1A3 as the focus of our subsequent research. The screening process and results are detailed in Figure 3.
Comments 3:
Figure 3F: While the claim that ALDH1A3 is a diagnostic marker is intriguing, it relies on expression data from a single cohort. To strengthen this claim, can the authors validate their findings using independent cohorts, such as patient-derived samples or publicly available datasets like GEO. Additionally, the authors could use ROC (Receiver Operating Characteristic) curve analysis to evaluate the sensitivity and specificity of ALDH1A3 as a marker.
Response 3: Thank you for your professional suggestions on our manuscript. We conducted a search of the GEO database and found that ALDH1A3 is upregulated in several prostate cancer GEO datasets, such as GSE32448, GSE46602, GSE200879, GSE55945, GSE60329, etc. However, when we applied the same selection criteria as in the manuscript, |logFC| ≥ 1 and adj. p < 0.05, at least three datasets—GSE80609, GSE133891, and GSE103512—met the criteria. We performed ROC curve analysis to assess the sensitivity and specificity of ALDH1A3 as a biomarker, and the results showed an AUC value of 0.887. These results support the potential of ALDH1A3 as a diagnostic marker.

Comments 4:
Beta-Galactosidase (SA-β-gal) staining is a widely used marker for cellular senescence, but it is not specific to senescence and can also indicate lysosomal activity in non-senescent cells. The authors’ conclusion of "accelerated senescence" solely based on SA-β-gal staining is not robust. The study should either include additional markers of senescence (e.g., p16, p21, IL6, IL8, IL1α, CXCL1, MMP9) to confirm the phenotype or reframe the findings as indicative of a "senescent-like phenotype." This adjustment is critical to ensure scientific accuracy.
Response 4: Thank you for your suggestion. Your input has helped make our manuscript more scientifically accurate. In response, we have revised the phrasing to "senescence-like phenotype. Additionally, in Figure 6D, we performed Western blot analysis to assess the expression of p16 following ALDH1A3 knockdown. The results showed a significant increase in p16 expression, further supporting the role of ALDH1A3 in cellular senescence.
Comments 5:
Figure Format: Several figures (e.g., Figures 3 and 6) appear to be inverted in the manuscript. This reduces the clarity of the data presentation. The authors should adjust the content and ensure that each figure is correctly displayed.
Response 5: Thank you for your valuable suggestion. We have re-uploaded Figures 3 and 6 in the manuscript and carefully reviewed all the figures to ensure their clarity and visibility. We believe these updates will improve the overall presentation of the data.
Comments 6:
Clarify and Enhance Figure Annotations: Many figures lack clear annotations, such as statistical significance levels or the rationale behind the specific comparisons made. Each figure should clearly highlight key findings for improved readability.
Response 6: Thank you for your suggestion. We have revised and double-checked the figure legends as per your request to ensure the content is clear and accurate.
Comments 7:
For Figure 4: Strengthen statistical analyses by Including sample sizes (n) in figure legends. Reporting effect sizes to complement p-values.
Response 7: Thank you for your valuable suggestion regarding Figure 4. We appreciate your recommendation to include sample sizes (n) in the figure legend, and we will ensure that this information is provided in the updated version of the manuscript. Additionally, we will report effect sizes to complement the p-values, as you suggested, to enhance the statistical analysis. These updates will improve the clarity and robustness of our data presentation.
Comments 8:
Clinical Implications: Expand on the therapeutic potential of targeting ALDH1A3. For instance, what are the challenges or risks of ALDH1A3 inhibition? Could this strategy be combined with existing treatments, such as senolytics or immune checkpoint inhibitors, to enhance efficacy? Are there any study limitation that the author can foresee?
Response 8: Thank you for your valuable suggestion. We have addressed the foreseeable limitations of our study in the final paragraph of the discussion and expanded on the potential and challenges of ALDH1A3-targeted therapy.
Comments 9:
The quality of English language in the manuscript is generally clear and professional, making the content understandable. However, there are occasional grammatical errors, awkward phrasing, and areas where the flow could be improved. Some sentences are long and complex, particularly in the discussion and results sections, which could benefit from being broken into shorter, more concise sentences. Certain phrases could also be refined for precision, such as replacing "dye deepening" with "increased staining intensity" to align with standard scientific terminology. Additionally, figure legends sometimes use informal or vague terms, which should be revised.
Response 9: Thank you for your valuable suggestion. We have reviewed and checked the language quality of the manuscript, simplifying and breaking down long and complex sentences in the Results and Discussion sections. "Dye deepening" has been replaced with "increased staining intensity." Additionally, we have re-checked and revised the figure legends.
Reviewer 2 Report
Comments and Suggestions for Authors
The manuscript entitled "ALDH1A3 regulates cellular senescence and senescence-associ-ated secretome in prostate cancer" is interesting and well-organized suggesting that ALDH1A3 could emerge as a new therapeutic target for prostate cancer. I have found the results of the manuscript very promising; however I have a major and a minor concern before its publication.
Major:
Authors need to evaluate the potential correlation between ALDH1A3 and senescence and secretome markers by performing Spearman's rank correlation coefficient (Spearman's rho) in publicly available prostate cancer clinical samples.
Minor:
Figure 3 and Figure 6 need to be replaced properly.
Author Response
Comments 1:
Authors need to evaluate the potential correlation between ALDH1A3 and senescence and secretome markers by performing Spearman's rank correlation coefficient (Spearman's rho) in publicly available prostate cancer clinical samples.
Response 1: Thank you for your valuable suggestion. Our study evaluated the correlation between ALDH1A3 and senescence and secretory marker genes such as CDKN1A, CDKN2A, IL6, IL1A, CXCL1, and MMP9. However, the results did not reveal any significant correlations. This may reflect the more indirect role of ALDH1A3 in the regulation of these markers, or its involvement in other, yet-to-be-identified biological mechanisms. Future research should focus on the potential complex role of ALDH1A3 in broader cancer biology processes and explore its potential functions in other signaling pathways.

Comments 2:
Figure 3 and Figure 6 need to be replaced properly.
Response 2: Thank you for your suggestion. We have re-uploaded Figure 3 and Figure 6 in the manuscript and have re-checked all the figures to ensure their clarity and visibility.
Reviewer 3 Report
Comments and Suggestions for Authors
In this work, the investigators have studied the impact of ALDH1A3 depletion in regulating senescence, and secretion of pro-inflammatory factors in prostate cancer cells. There are several concerns with the study and the data presented in its current form.
1. The reason for selecting prostate cancer as the model to study is not explained at all in the Introduction or Abstract, and is only explained as part of Figure 3. They must demonstrate that ALDH1A3 exhibits similar roles in other cancer as well, if they do not highlight the reason for selecting prostate cancer foremost
2. The more important question is - which pathways and signaling process are impacted at the early time points and how many of these changes are sustained through the later time point? In other words, do the molecular alterations induced soon after radiotherapy modify or regulate delayed onset reactions in the cancer cells? By not examining the data with this question in mind, the investigators have neglected to review a vital set of information
3. The investigators have examined the extent of immune infiltration in prostate cancer samples with different levels of expression of ALDH1A3 and highlighted putative immune cell types. The more important question is that in the context of radiotherapy, how are these immune cells impacted functionally? To address that question, the investigators need to set up co-culture assays of cancer cells and immune cells and irradiate these cells together. No such experiment has been conducted, and one does not know how immune cells will function in the context of radiotherapy of the prostate cancer cells.
4. The knock-out studies are not convincing, especially with sgRNA-1. What is the level of ALDH1A3 after the knock-out? Both cell proliferation wound closure assays exhibit proliferative and migratory abilities of the irradiated cancer cells. The authors need to show the p values of the differences recorded, because visually the difference looks to be significant between control and knock-out cells.
MINOR COMMENT
Figure 6 is flipped in the manuscript. Please re-format.
Author Response
Comments 1:
The reason for selecting prostate cancer as the model to study is not explained at all in the Introduction or Abstract, and is only explained as part of Figure 3. They must demonstrate that ALDH1A3 exhibits similar roles in other cancer as well, if they do not highlight the reason for selecting prostate cancer foremost
Response 1: Thank you for your valuable suggestion. We have highlighted the reasons for choosing prostate cancer as the study model in the first paragraph of the introduction based on your suggestion.
Comments 2:
The more important question is - which pathways and signaling process are impacted at the early time points and how many of these changes are sustained through the later time point? In other words, do the molecular alterations induced soon after radiotherapy modify or regulate delayed onset reactions in the cancer cells? By not examining the data with this question in mind, the investigators have neglected to review a vital set of information
Response 2: Thank you for your valuable comments. Our current study did focus on analyzing changes in gene and protein expression at different time points after radiotherapy, especially comparing early (e.g. 24 hours) and late (e.g. 96 hours) responses. We observed that some of the early molecular alterations, such as cell cycle arrest and DNA repair responses, were still present at later time points, such as after 7 days. However, a deeper understanding of how these early changes modulate the delayed response of cancer cells, and further exploration of whether these early changes affect tumor cell aging, immune escape, or other radiotherapy-related delayed responses in the long term, are our follow-up research plans.
Comments 3:
The investigators have examined the extent of immune infiltration in prostate cancer samples with different levels of expression of ALDH1A3 and highlighted putative immune cell types. The more important question is that in the context of radiotherapy, how are these immune cells impacted functionally? To address that question, the investigators need to set up co-culture assays of cancer cells and immune cells and irradiate these cells together. No such experiment has been conducted, and one does not know how immune cells will function in the context of radiotherapy of the prostate cancer cells.
Response 3: Thank you for your valuable suggestions to our study. We understand your demand for experiments on the function of immune cells in the context of radiotherapy, especially the co-culture experiments of cancer cells and immune cells, which will provide an important experimental basis for us to further understand the role of immune cells in radiotherapy of prostate cancer.
At present, due to the limitations of experimental conditions and resources, we are unable to carry out this experiment. However, we have preliminarily investigated the relationship between ALDH1A3 and immune cell infiltration by analyzing published data sets and literature, and have obtained relevant analytical results. Although we are currently unable to experimentally verify the functional changes of immune cells during radiotherapy, we plan to further design and implement relevant experiments in future studies to complement this part of the study.
Comments 4:
The knock-out studies are not convincing, especially with sgRNA-1. What is the level of ALDH1A3 after the knock-out? Both cell proliferation wound closure assays exhibit proliferative and migratory abilities of the irradiated cancer cells. The authors need to show the p values of the differences recorded, because visually the difference looks to be significant between control and knock-out cells.
Response 4: Thank you for your valuable suggestions to our study. Regarding the knockout experiment, we performed grayscale analysis on the ALDH1A3 expression levels after knockout. The results show that the ALDH1A3 expression in the knockout group is significantly lower than in the control group. We have added the p-value in the figure legend.

Comments 5:
Figure 6 is flipped in the manuscript. Please re-format.
Response 5: Thanks for your suggestion, we re-upload and check the picture again.
Round 2
Reviewer 1 Report
Comments and Suggestions for Authors
The manuscript entitle "LDH1A3 regulates cellular senescence and senescence-associated secretome in prostate cancer" explores the role of ALDH1A3 in prostate cancer, with a particular focus on its impact on cellular senescence and the senescence-associated secretory phenotype (SASP). By integrating transcriptomic and proteomic data, the authors demonstrate that ALDH1A3 knockdown accelerates a senescence-like phenotype but reduces SASP-related pro-inflammatory factors via the cGAS-STING pathway. Dear Authors, thank you for including my suggestion to your manuscript. The revised version have improved now the statistics, an expanded introduction and discussion on the findings. I do recommend for publications.
Best wishes
Author Response
Comments :
The manuscript entitle "LDH1A3 regulates cellular senescence and senescence-associated secretome in prostate cancer" explores the role of ALDH1A3 in prostate cancer, with a particular focus on its impact on cellular senescence and the senescence-associated secretory phenotype (SASP). By integrating transcriptomic and proteomic data, the authors demonstrate that ALDH1A3 knockdown accelerates a senescence-like phenotype but reduces SASP-related pro-inflammatory factors via the cGAS-STING pathway. Dear Authors, thank you for including my suggestion to your manuscript. The revised version have improved now the statistics, an expanded introduction and discussion on the findings. I do recommend for publications.
Reply:
Thank you for your positive feedback and for recommending our manuscript for publication. We greatly appreciate your valuable suggestions, which have helped us improve the statistical analysis, as well as expand the introduction and discussion. Your insights have been instrumental in refining our work.
Reviewer 2 Report
Comments and Suggestions for Authors
Dear Authors!
I would like to thank you for informing me about the ALDH1A3 potential correlation with senescence and secretory marker genes. Although, ALDH1A3 did not reveal any significant correlation with these markers; you should include this figure in the supplementary materials.
Author Response
Comment:
I would like to thank you for informing me about the ALDH1A3 potential correlation with senescence and secretory marker genes. Although, ALDH1A3 did not reveal any significant correlation with these markers; you should include this figure in the supplementary materials.
Response:
We sincerely appreciate your valuable feedback and your suggestion regarding the correlation analysis of ALDH1A3 with senescence and secretory marker genes. As per your suggestion, we have added the corresponding description in lines 467 to 472 and included the figure in the supplementary materials.
Thank you for your insightful comments, which have helped improve the clarity and completeness of our manuscript.